# Neuromechanisms and subjective experiences during human-dog interactions: Assessing motivation and mental state in a randomized, controlled trial

Fabio Carbone[1,2]*, Eve-Yaël Gerber[1☉], Camille Rérat[1☉], Jan Hattendorf[3], Karin Hediger[1,2,4]

**1** Faculty of Psychology, University of Basel, Basel, Switzerland, **2** REHAB Basel, Basel, Switzerland, **3** Department of Epidemiology and Public Health, Swiss Tropical and Public Health Institute, Allschwil, Switzerland, **4** Faculty of Behavioral Sciences and Psychology, University of Lucerne, Lucerne, Switzerland

☉ These authors contributed equally to this work.
* Fabio.carbone@unibas.ch

## Abstract

Animal Assisted Interventions (AAIs) have been shown to have several effects in humans but the underlying cerebral mechanisms are still widely unknown. This research explored the neurological aspects of human–animal interactions. Specifically, we focused on frontal alpha asymmetry (FAA), a feature indicating differences in alpha power between the left and right frontal cortex, which is recognized as a correlate of approach motivation and positive affect. Twenty-nine healthy adults participated in this study, in which we used electroencephalography to measure their brain activity. The study comprised five phases: baseline measurements, interaction with a real dog, interaction with a replica dog, interaction with a plant, and a neutral phase. Participants had both physical and visual contact with the real dog, the replica and the plant, and the procedure was repeated three times for each participant. We also assessed participants' subjective experiences of mental states and intrinsic motivation through the Multidimensional Well-Being and the Intrinsic Motivation Inventory questionnaires. The objective measurements of motivation and positive affect through FAA did not show a significant difference between interactions with a real dog and control conditions, but the subjective assessments differed. Participants reported significantly higher motivation and a more positive state of mind after interacting with a real dog compared to the control conditions. These results could be considered in therapeutic settings when determining whether to incorporate an animal into a treatment plan. In summary, this study highlights the complexity of human–animal interactions (HAI) and shows an intricate interplay between objective and subjective measurements. Our findings emphasize the importance of considering both neural

**Data availability statement:** All data and codes are available on the OSF platform: https://osf.io/tz26c/?view_only=d673459b9ad-64b6986173aa97d129b21 or upon reasonable requests.

**Funding:** Funding for this project was provided by the Swiss National Science Foundation (SNSF) through the Eccellenza grant PCEFP1_194591 / 1. The funders had no role in study design, data collection and analysis, decision to publish, or preparation of the manuscript.

**Competing interests:** The authors have declared that no competing interests exist.

markers and subjective experiences for understanding the nuanced mechanisms involved in the meaningful connections humans have with animals.

## Introduction

Human–animal interactions (HAIs) have been shown to improve mood, reduce stress, and increase motivation [1–4]. These effects are often attributed to the emotional and motivational benefits of interacting with animals. However, while Animal-Assisted Interventions (AAIs) are gaining popularity in clinical and non-clinical settings, the underlying neurological mechanisms remain poorly understood. A clearer understanding of these mechanisms is essential to optimize and tailor interventions involving animals.

Animals can elicit emotional responses in humans that may enhance motivation and positive affect. Pet ownership has been linked to psychological benefits [5], and engaging in specific activities with animals, such as walking or playing with a dog, has been shown to improve mood and reduce stress [6–8]. In therapeutic contexts, animals can enhance clients' engagement and motivation to participate in treatment [9–12]. In pediatric settings, animal contact has been shown to reduce distress and enhance communication and relaxation during therapy [13,14].

Despite these promising outcomes, research on the neural basis of such effects is still limited. Previous neuroimaging studies have explored brain responses to HAIs using methods such as functional magnetic resonance imaging (fMRI), functional near-infrared spectroscopy (fNIRS), and electroencephalography (EEG). These studies suggest that interacting with or even viewing animals can activate regions associated with emotion and attention, such as the prefrontal cortex and amygdala [15–20]. For example, EEG studies have shown increased beta and alpha activity during interaction with animals, which may reflect heightened attention and relaxation, respectively [8,21–24]. However, these studies vary widely in methods, populations (e.g., children, elderly, clinical), and contexts (e.g., post-surgical recovery, interactions with healthy participants), making it difficult to draw general conclusions about the effects of HAI on brain activity in healthy adults.

One promising avenue for understanding the neural basis of motivation and affect in HAIs is frontal alpha asymmetry (FAA), a well-established EEG marker associated with approach motivation and positive affect. Lower alpha power (which reflects increased cortical activity) in the left relative to the right frontal cortex is linked to greater approach motivation and positive emotional states [25–27]. Although FAA has been widely studied in emotion research, it has not yet been systematically applied to HAIs in healthy populations.

The present study investigates whether the presence of and interaction with a real dog elicits changes in FAA in healthy adults compared to two control conditions (a replica dog and a plant). We used EEG to assess alpha power in the prefrontal and frontal cortex during interactions with each stimulus. Moreover, we measured the participant's subjective experienced motivation and current mental state using the Intrinsic Motivation Inventory (IMI) [28] and the Multidimensional Wellbeing Questionnaire (MDWB) [29].

We hypothesized that interacting with a real dog would result in greater left-sided frontal activation (i.e., increased FAA) compared to the control conditions, reflecting enhanced approach motivation and positive affect. This effect would be most pronounced in the dog condition relative to the replica dog and plant. Regarding subjective motivation and wellbeing, we also hypothesized that they will be higher in the dog condition in comparison with the two controls.

To further explore the neural dynamics of HAI, we also conducted an exploratory analysis replicating the frequency spectrum analysis from Yoo et al. [8] though those results are not reported in the present publication.

The study aims to close the gap in understanding the effects of HAI approach motivation and positive affect in healthy adults, using both physiological (FAA) and subjective (IMI, MDWB) measures to ensure a multidimensional perspective.

## Material and methods

### Study population

Recruitment was conducted via online advertisements describing the study design, the intervention, and the duties of the participants. The ad was published on a dedicated website accessible by university students and general public. We aimed for a gender-balanced sample. The inclusion criteria were an age of 18 or over and the ability to give written consent. We excluded people who declared a fear of dogs, allergies to dogs, or any acute or chronic diseases (e.g., chronic pain, hypertension, heart disease, renal disease, liver disease, diabetes). We also excluded anyone who was currently on medications (e.g., psychoactive medications, narcotics, analgesics) or was in psychological or psychiatric treatment at the time of the recruiting, as well as anyone who declared current or regular drug consumption (THC 24h before visit, cocaine, heroin, etc.). We obtained written informed consent from every participant before the study started. The recruitment took place between April 2023 and July 2023. The participants received a compensation of 100 Swiss francs for their full participation.

### Determination of sample size and randomization

Based on the results published by Barcelona et al. [30], we estimated that 24 participants in total would be required to detect an effect size of 0.06 in the primary analysis with 80% power at a 95% significance level. Even if the estimated effect size is considered small, it is in the range of what is expected for an FAA analysis measuring the effect of a psychological intervention [31]. To have a balanced sample and to include a small safety margin, we increased the anticipated total sample size to 30. The randomization of the 90 sessions was performed before the recruitment of the first participant using an in-house R script. Each three sessions were allocated to the 30 ID numbers, and each participant received a random ID number as we went along with recruitment.

### Procedure

**Screening procedure.** Participants received the informed consent as well as the Animal Attitude Scale (AAS) questionnaire [32,33] and the Coleman Dog Attitude Scale (C-DAS) [34] to determine their attitude toward animals and dogs. The AAS is composed of 20 items about animal welfare. Participants are asked to indicate the extent to which they agree with statements on a scale from 1 (strongly disagree) to 5 (strongly agree). The total score is then calculated with a maximum possible score of 100, a higher score indicating attitudes in favor of animal welfare. The C-DAS is composed of 24 statements about attitudes toward dogs. It also asks participants to indicate the extent to which they agree with statements on a scale from 1 (strongly disagree) to 5 (strongly agree). The total score is then calculated with a maximum possible score of 120, a higher score indicating a positive attitude toward dogs. After the questionnaires, the candidates were invited for an online screening interview and were informed about all aspects of the study. The inclusion and exclusion criteria were checked by the experimenter during the screening interview. Due to the nature of the intervention, no medical screening was needed, and the exclusion and inclusion criteria were controlled on a self-reporting basis.

**Experimental procedure.** The study was a within-subject randomized controlled trial with repeated measurements. All participants underwent the same protocol involving three independent EEG-measurement sessions spaced one week apart and always taking place at the same time of day for any given participant. Each session exposed participants to three distinct conditions: real dog, replica dog, and plant. The order of the three conditions was randomized for every session to control for potential sequence effects. Each session started with a baseline assessment of the participant's current mental state using the MDWB questionnaire and a baseline EEG measurement, during which participants focused on a cross on a wall for 5 minutes. Subsequently, the EEG measurement for the first condition lasted 5 minutes and was immediately followed by the IMI questionnaire and another MDWB assessment. This procedure was repeated for the second and third conditions. A neutral phase, similar to the initial baseline, was applied at the end of the session, providing a comprehensive and balanced design for investigating the impact of different stimuli on brain activity and subjective experiences (Fig 1).

## Study intervention

The intervention was the presence of a real dog (HAI). The term *presence* describes a dog in the room including physical contact between the dog and the participant whenever feasible. This contact was established by instructing the dog to lie down beside the participant and directing the participant to gently pet the dog with their right hand. Participants were seated on a low chair to facilitate contact with the dog (Fig 2). The dogs involved in the study underwent specific training for this task. The duration of the interaction was systematically monitored by a study-team member using a scale ranging from 0 to 5, where 0 indicated no physical contact at all with the participant, and 5 signified continuous contact throughout the entire 5-minute recording.

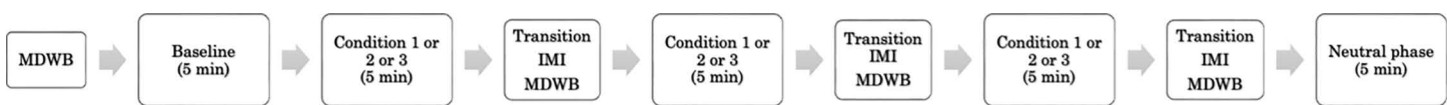

**Fig 1. Sequence of measurements in a session.** The sequence was repeated three times, one week apart. The order of the conditions (dog, dog replica, or plant) was randomized for each session. MDWB = Multidimensional Well-Being questionnaire; IMI = Intrinsic Motivation Inventory questionnaire.

A B C

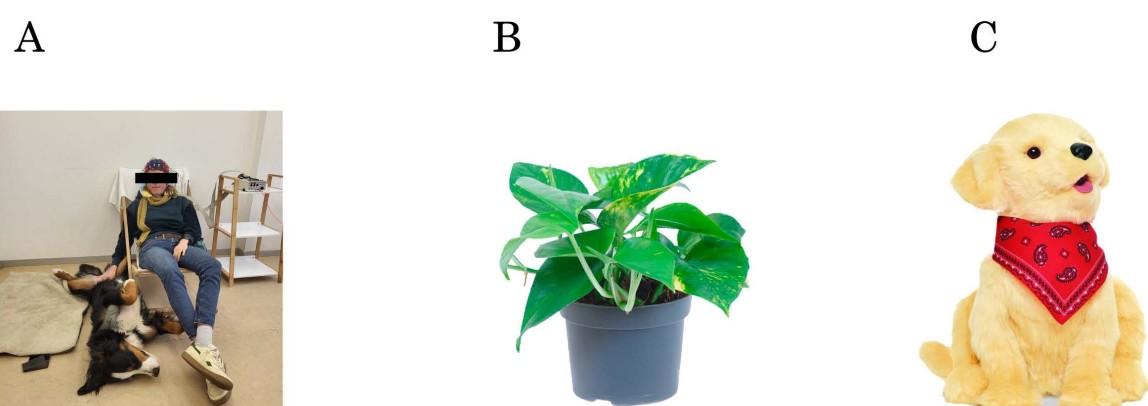

**Fig 2. Experimental conditions.** (A) Dog condition. A similar recording setup as shown in the picture was used for all the conditions. (B) Plant condition (*Epipremnum aureum*). (C) Replica-dog condition *Medor* (Joy for AI™, © 2020 Ageless Innovation).

**Dogs.** The dogs, having been trained for such interactions, were accustomed to working with various individuals in the context of HAIs and were familiarized with the room, materials, and study staff. To ensure their well-being and to manage their workload, each dog participated in a maximum of three sessions per day lasting approximately 20 minutes in total, and they engaged in the intervention on a maximum of 2 days per week. Three dogs were involved in the study: a 3-year-old male flat-coated retriever, a 3-year-old female Bernese Mountain dog, and a 3-year-old mixed breed male. The same dog was always assigned to a single participant.

**Replica dog.** As a primary control condition, we introduced a therapeutic replica dog named Medor (Joy for All™, © 2020 Ageless Innovation). This lifelike robotic pet companion has fur, can bark, replicates a heartbeat, and responds to touch and voice interactions. Frequently employed in home care and nursing homes, the replica dog closely mimics the appearance and interactive features of a real dog. Like the real dog, Medor was placed next to the participant, and instructions were provided akin to those for the real-dog interaction. Participants were asked to touch and pet the replica dog using their right hand.

**Plant.** As an additional control condition, we incorporated a real plant (*Epipremnum aureum*). The selection of a plant aimed to gauge engagement with an everyday object that diverges significantly from an animal in appearance. Notably, previous research has explored the influence of a plant's presence on electrical activity in the frontal brain using precisely this plant species [35]. Participants received instructions parallel to those given for the real-dog interaction and were asked to touch the plant using their right hand.

## Biological measures

**EEG-data acquisition.** The EEG device employed in this study was composed of 64 pin-type active AgCl electrodes produced by BioSemi BV, Amsterdam, following the international 10–10 system. To ensure accurate placement, the head cap was centered on participants' heads by positioning Cz at the intersection of the anion–nasion and ear-to-ear axes. Data were captured using the software ActiView (version 9.02, BioSemi BV, Amsterdam) at a sampling rate of 2048 Hz. The electrodes were installed as per the manufacturer's instructions, maintaining an electrode offset within ±40 mV. All conditions, including the baseline and the neutral phase, were recorded in a single session. The recording was momentarily paused during transitions to prevent unnecessary data acquisition. Specific markers denoted the start and end of each condition.

**EEG-data preprocessing.** After recording, the data underwent editing and cropping using the software ActiRead and ActiTool (version 9.02, BioSemi BV, Amsterdam), retaining the segments corresponding to the 5 minutes of exposure for each condition. A total of 435 recordings were successfully acquired; they formed the basis for the subsequent analysis. The data were preprocessed in MATLAB (version 9.12.0.2009381 (R2022a) update 4) using the toolbox EEGLAB [36]. The data were imported using the BIOSIG plugin (version 3.8.0; [37]). An average reference was computed, and the data were high-pass filtered at 0.5 Hz and low-pass filtered at 75 Hz with basic FIR filters. Major movement artifacts and defect channels were manually removed. Independent component analysis (ICA) was run to identify artifact components in the signal. An automated artifact rejection was performed using the EEGLAB plugin ADJUST (version 1.1.1; [38]), which has been shown to be appropriate for computing FAA [39].

**FAA calculation.** FAA was computed on three pairs of electrodes: F3–F4, F7–F8, Fp1–Fp2. The EEGLAB plugin FAA [40] was used to compute FAA on a frequency range 8–13 Hz. The plugin uses the spectrogram function to return short-time Fourier transformations. The FAA-calculation parameter was a 1.5-second length of Hamming window with 50% overlap. The FAA formula used in the toolbox was modified to match the standard FAA formula: $ln(right\ alpha\ power) - ln(left\ alpha\ power)$ [41].

## Psychological Measures

**Motivation.** Subjective motivation was measured with an IMI questionnaire [28] adapted to assess motivation after an activity [42]. The authors describe their questionnaire as "a multidimensional measurement device intended to

assess participants' subjective experience related to a target activity in laboratory experiments." The version used in this study consists of a 25-item survey, and each item belongs to one of the three subscales value, interest, and choice. They respectively represent the value/usefulness given to an activity, the interest/enjoyment felt for an activity, and the perceived choice to take part in an activity. For each item, participants are asked to indicate how true the statement is on a scale from 1 (not true at all) to 7 (very true). The subscale scores are calculated by averaging the score of their corresponding items, thus making a range from 1 to 7 for each subscale. The interest/enjoyment subscale is considered to be a self-reported measure of intrinsic motivation. The perceived choice is considered a positive predictor of both self-reported and behavioral measures of intrinsic motivation. The value/usefulness subscale assesses the extent to which individuals internalize and autonomously regulate their engagement in activities based on their perception of the activities as personally valuable or useful.

**Multidimensional mental state.** Subjective current mental state was assessed with the short versions A and B of the MDWB questionnaire [29]. It consists of two matched 12-item surveys. The items are adjectives chosen to describe the mood of participants in the moment they are filling out the questionnaire. Both versions use different words, but the outcome is comparable. For each item, participants give an answer on a scale from 1 (not at all) to 5 (very). The questionnaire has three subscales: good–bad mood (GB), wakefulness–tiredness (WT), and calmness–restlessness (CR). The scores of each subscale are summed and range between 8 and 40. A high score in the GB scale indicates a positive mood and reflects a person that feels good, happy, and satisfied. On the other hand, a low score indicates that a person feels unwell, in a bad mood, and dissatisfied. A high score on the WT scale results if people are awake and rested, feeling fresh and alert. On the contrary, a low score indicates feeling tired, sleepy, and listless. Finally, a high value on the CR scale reflects a respondent who feels rather calm and relaxed. On the opposite, a low score reflects a tense, agitated, nervous, and restless person.

## Statistical analysis

The primary outcomes were the approach motivation and the positive affect measurements as operationalized via FAA scores. The secondary outcomes were intrinsic motivation and state of mind as measured by IMI and MDWB, respectively. All analyses were performed on R (version 4.3.0) using the packages lme4 [43] for the linear mixed-effect model (LMM) and emmeans (version 1.8.7; [44]) for the pairwise comparisons and effect-size estimations.

**FAA.** The analysis was performed on the hemispheric asymmetry of the averaged alpha power in the prefrontal and frontal cortex during the intervention and control conditions using an LMM. The FAA score was the response variable. The conditions and the sessions were fixed effects, and the individual participants were defined as random effects. AAS scores and C-DAS were additional independent variables included in the model. The first model was:

$$Score[ij] \sim Constant + Condition[ij] + Session[ij] + AAS[ij] + CDAS[ij] + ID[i]$$

The model was run in the same way for each of the three pairs of electrodes Fp1–Fp2, F3–F4, and F7–F8. We visually checked the normality (Q–Q plot, histogram of residuals), linearity, and homoscedasticity (residuals vs. fitted plot). The significance level was set at .05. When the session variable, the AAS scores, or the CDAS scores were nonsignificant, they were removed from the model. Pairwise comparisons between each condition were calculated, as was a Cohen's $d$ effect size for each pairwise comparison. Multitesting for each pairwise comparison was corrected with the Bonferroni method.

**IMI.** The analysis was performed individually on the three subscales interest, value, and choice using an LMM. The IMI scores were the response variables. The conditions and the sessions were fixed effects, and the individual participants were defined as random effects. The condition dog was set as the reference level since the design of the questionnaire—that is, the evaluation of intrinsic motivation after a task—made the acquisition of a baseline measurement impossible. We visually checked the normality (Q–Q plot, histogram of residuals), linearity, and homoscedasticity (residuals vs. fitted plot).

The significance level was set at .05. Pairwise comparisons between each condition were calculated, as was a Cohen's *d* effect size for each pairwise comparison. Multitesting for each pairwise comparison was corrected with the Bonferroni method.

**MDWB.**  The analysis was performed individually on the three subscales GB, CR, and WT using an LMM. The MDWB scores were the response variables. The conditions and the sessions were fixed effects, and the individual participants were defined as random effects. We visually checked the normality (Q–Q plot, histogram of residuals), linearity, and homoscedasticity (residuals vs. fitted plot). The significance level was set at .05. Pairwise comparisons between each condition were calculated, as was a Cohen's *d* effect size for each pairwise comparison. Multitesting for each pairwise comparison was corrected with the Bonferroni method.

**Correlations between FAA and IMI/MDWB.**  Pearson correlations for each pair of electrodes were calculated between the FAA scores of each condition and the subscale scores of the IMI questionnaire. Similarly, the correlations for each pair of electrodes were calculated between the FAA scores of each condition and the scores of the three subscales of the MDWB questionnaire. The results are shown in the supplementary material (S4–S6 Figs).

**Correlations between FAA and interaction time.**  Pearson correlations for each pair of electrodes were calculated between the FAA scores of the dog condition and the duration of the physical interaction between the participant and the dog. The results are shown in the supplementary material (S10 Table).

## Ethical statement

The study protocol, information sheet and informed consent and all documents handed over to participants were approved by the local ethics committee, the Ethics Commission Northwest and Central Switzerland (Project ID 2023_00206) and was registered at clinicaltrials.gov (NCT05837546). All sessions were conducted according to the guidelines of the International Association for Human–Animal Interaction Organizations (IAHAIO) and the Helsinki guidelines [45,46].

## Results

### Participant characteristics

Thirty young healthy adults were recruited for the study (*mean age = 28.07 years ± SD 10.11 years; 16 females, 14 males*). One participant was excluded from the study after screening for noncompliance with the study protocol. Twenty-nine participants (*mean age = 28.07 years ± SD 10.3 years, 15 females, 14 males*) were included in the data analysis (Table 1).

**FAA.**  A descriptive analysis of all the electrode pairs can be found in Table 2 and a boxplot of the distribution of FAA scores for each pair of electrodes in Fig 3. Individual distributions for each electrode pair in all conditions and sessions are shown in supplementary material (S1–S3 Figs).

**Fp1–Fp2.**  The LMM was fitted to examine the relationship between the FAA score as the dependent variable for the electrode pair Fp1–Fp2 and the fixed effect of the condition as the categorical predictor with four levels: dog, replica, plant, and neutral; the reference condition was the baseline. Eighty-seven recordings of each condition were used in the analysis. The model also accounted for the random effects of the variable ID, which represented the participants. The independent variables session (estimate = 0.017, *p* = 0.464), AAS scores (estimate = 0.002, *p* = 0.357), and C-DAS scores (estimate = −0.001, *p* = 0.730) did not contribute significantly to the first model and were then removed. The final model was specified as:

$$Score[ij] \sim Condition[ij] + ID[i]$$

The intercept was estimated to be non-significant, with a slight positive value. For the dog condition, the analysis showed no significant difference in FAA scores compared to the baseline. Similarly, the plant condition did not result in any notable change in FAA scores. The replica condition also demonstrated no significant variation in FAA scores from the baseline.

Finally, the neutral phase indicated no meaningful difference in FAA scores compared to the baseline condition. The outcomes of the model are shown in Table 3.

No pairwise comparison between the different conditions was found significantly different. Statistics can be found in the supplementary materials (S1 Table).

**F3–F4.** The analysis involved fitting an LMM to explore the association between the FAA score for the electrode pair F3–F4 and the fixed effects related to different experimental conditions. These conditions—namely, dog, replica, plant, and neutral—were compared to the baseline condition. Eighty-seven recordings of each condition were used in the analysis, except for the dog condition, for which 86 recordings were used, one being excluded due to technical issues. The model considered the random effects associated with participant ID. Subsequently, noncontributory independent variables (AAS and C-DAS scores) were excluded. The order of the sessions was significant, so it was kept in the final model:

$$Score[ij] \sim Condition[ij] + Session[ij] + ID[i]$$

**Table 1. Participant characteristics. Gender, age, and attitude toward animals and dogs.**

| ID | Gender | Age (year) | AAS score | C-DAS score |
|---|---|---|---|---|
| 1 | m | 30 | 63 | 87 |
| 2 | f | 45 | 84 | 79 |
| 3 | m | 20 | 81 | 90 |
| 4 | f | 24 | 82 | 100 |
| 5 | f | 25 | 76 | 95 |
| 6 | m | 28 | 81 | 80 |
| 7 | f | 23 | 81 | 103 |
| 8 | m | 42 | 74 | 73 |
| 9 | m | 21 | 67 | 78 |
| 10 | f | 20 | 89 | 97 |
| 11* | f | 28 | 81 | 68 |
| 12 | f | 24 | 86 | 89 |
| 13 | f | 27 | 90 | 92 |
| 14 | f | 20 | 84 | 102 |
| 15 | m | 25 | 80 | 97 |
| 16 | f | 21 | 73 | 78 |
| 17 | f | 28 | 85 | 105 |
| 18 | m | 27 | 72 | 87 |
| 19 | f | 23 | 65 | 75 |
| 20 | m | 62 | 93 | 80 |
| 21 | m | 27 | 84 | 106 |
| 22 | m | 27 | 43 | 82 |
| 23 | f | 19 | 67 | 96 |
| 24 | m | 32 | 86 | 80 |
| 25 | m | 22 | 77 | 89 |
| 26 | f | 53 | 83 | 101 |
| 27 | m | 19 | 68 | 103 |
| 28 | m | 23 | 69 | 76 |
| 29 | f | 23 | 89 | 102 |
| 30 | f | 34 | 69 | 77 |

*Note. ID = participant number; AAS = Animal Attitude Scale; C-DAS = Coleman Dog Attitude Scale; m = male; f = female; \*participant was excluded.*

**Table 2. Descriptive analysis of the three electrode pairs for each condition.**

| Electrode pair | Condition | Mean | Median | SD | Min | Max | Count |
|---|---|---|---|---|---|---|---|
| Fp1–Fp2 | Baseline1 | 0.077 | 0.032 | 0.368 | −0.974 | 1.484 | 87 |
| | Dog | 0.020 | 0.080 | 0.538 | −1.278 | 1.207 | 87 |
| | Neutral | 0.052 | 0.001 | 0.290 | −0.701 | 1.160 | 87 |
| | Plant | 0.014 | −0.033 | 0.325 | −0.903 | 0.783 | 87 |
| | Replica | −0.006 | 0.019 | 0.440 | −1.558 | 0.996 | 87 |
| F3–F4 | Baseline1 | 0.109 | 0.058 | 0.326 | −0.600 | 0.971 | 87 |
| | Dog | 0.148 | 0.203 | 0.634 | −1.622 | 2.166 | 86 |
| | Neutral | 0.038 | 0.000 | 0.299 | −0.815 | 0.885 | 87 |
| | Plant | 0.169 | 0.118 | 0.463 | −0.745 | 2.011 | 87 |
| | Replica | 0.181 | 0.089 | 0.493 | −0.949 | 1.489 | 87 |
| F7–F8 | Baseline1 | −0.076 | −0.069 | 0.493 | −1.886 | 1.219 | 86 |
| | Dog | 0.034 | 0.014 | 0.566 | −1.125 | 1.332 | 86 |
| | Neutral | −0.084 | −0.078 | 0.393 | −1.073 | 1.224 | 87 |
| | Plant | −0.137 | −0.027 | 0.460 | −1.075 | 1.055 | 85 |
| | Replica | −0.027 | −0.017 | 0.557 | −1.187 | 1.256 | 85 |

Note. Mean, median, standard deviation (SD), minimum and maximum of frontal alpha asymmetry scores for each condition.

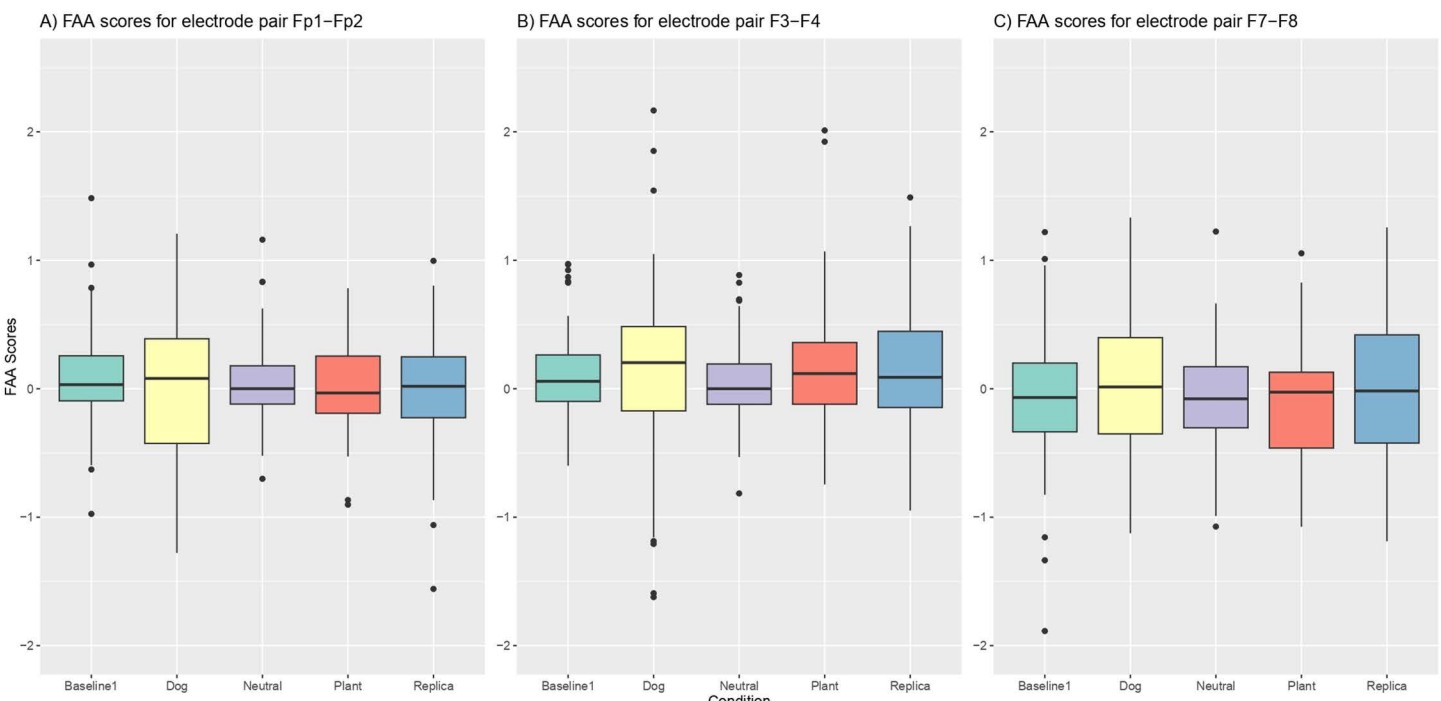

**Fig 3. Boxplot of frontal alpha asymmetry (FAA).** Left panel: electrode pair Fp1–Fp2. Middle panel: Electrode pair F3–F4. Right panel: Electrode pair F7–F8. Scores across five experimental conditions: Baseline 1, Neutral, Dog, Replica, and Plant. Each box represents the interquartile range (IQR) with the median indicated by the horizontal line, and whiskers extending to 1.5 × IQR. N = 29, repeated measurements over three sessions.

**Table 3. Results for condition effects for electrodes pair Fp1-Fp2.**

| Predictors | Estimate | SE | CI_Lower | CI_Upper | p_value |
|---|---|---|---|---|---|
| Intercept | −0.075 | 0.061 | −0.194 | 0.043 | 0.218 |
| Condition Dog | 0.110 | 0.070 | −0.028 | 0.248 | 0.119 |
| Condition Neutral | −0.009 | 0.070 | −0.146 | 0.128 | 0.896 |
| Condition Plant | −0.060 | 0.071 | −0.198 | 0.078 | 0.393 |
| Condition Replica | 0.050 | 0.071 | −0.088 | 0.188 | 0.477 |

*Note. Estimates, standard errors (SE), confidence intervals (CI), and p-values are presented for each predictor. Confidence intervals represent the 95% confidence level.*

The analysis revealed an estimated intercept that did not significantly deviate from zero. The dog, plant, replica, and neutral conditions all failed to show statistically significant effects on the FAA score. However, the session variable had a statistically significant impact on the FAA score, indicating a meaningful effect. The outcomes of the model are shown in Table 4.

None of the pairwise comparisons between the different conditions were found to be significantly different. Statistics can be found in the supplementary materials (S2 Table).

**F7–F8.** An LMM was employed to investigate the association between the dependent variable of the FAA score, measured for the electrode pair F7–F8, and the condition as the categorical predictor. Condition had four levels: dog, replica, plant, and neutral, with the baseline condition as the reference. Eighty-seven recordings of the neutral condition, 86 of the baseline condition, 86 of the dog condition, 85 of the plant condition, and 85 of the replica condition were used in the analysis. The missing recordings were excluded due to technical issues or excessive artifacts. The model incorporated random effects for the variable ID, which represented individual participants. The independent variables session (estimate = 0.009, $p = 0.733$), AAS scores (estimate = −0.003, $p = 0.571$) and C-DAS scores (estimate = 0.001, $p = 0.908$) did not significantly contribute to the first model and were then removed. The final model was specified as:

$$Score[ij] \sim Condition[ij] + ID[i]$$

The LMM analysis revealed an intercept that did not significantly deviate from zero. The dog, plant, replica, and neutral conditions all showed nonsignificant effects on FAA scores, indicating no meaningful differences from the baseline. The outcomes of the model are shown in Table 5.

None of the pairwise comparisons between the different conditions were found to be significantly different. Detailed statistics can be found in the supplementary materials (S3 Table).

**Table 4. Results for condition effects for electrodes pair F3-F4.**

| Predictors | Estimate | SE | CI_Lower | CI_Upper | p_value |
|---|---|---|---|---|---|
| (Intercept) | −0.022 | 0.075 | −0.169 | 0.125 | 0.767 |
| Condition Dog | 0.038 | 0.062 | −0.083 | 0.160 | 0.540 |
| Condition Neutral | −0.071 | 0.062 | −0.192 | 0.050 | 0.249 |
| Condition Plant | 0.060 | 0.062 | −0.061 | 0.181 | 0.335 |
| Condition Replica | 0.071 | 0.062 | −0.050 | 0.192 | 0.250 |
| Session | 0.066 | 0.024 | 0.019 | 0.113 | **0.006** |

*Note. Estimates, standard errors (SE), confidence intervals (CI), and p-values are presented for each predictor. Confidence intervals represent the 95% confidence level.*

**Table 5. Results for Condition Effects for Electrodes Pair F7-F8.**

| Predictors | Estimate | SE | CI_Lower | CI_Upper | p_value |
|---|---|---|---|---|---|
| Intercept | −0.075 | 0.061 | −0.194 | 0.043 | 0.218 |
| Condition Dog | 0.110 | 0.070 | −0.028 | 0.248 | 0.119 |
| Condition Neutral | −0.009 | 0.070 | −0.146 | 0.128 | 0.896 |
| Condition Plant | −0.060 | 0.071 | −0.198 | 0.078 | 0.393 |
| Condition Replica | 0.050 | 0.071 | −0.088 | 0.188 | 0.477 |

*Note. Estimates, standard errors (SE), confidence intervals (CI), and p-values are presented for each predictor. Confidence intervals represent the 95% confidence level.*

**IMI. Descriptive IMI.** A descriptive analysis of the IMI questionnaire can be found in Table 6 and a boxplot of the distribution of IMI scores for each subscale in Fig 4.

**Interest.** An LMM was employed to investigate the relationship between the dependent variable of the IMI score for the interest subscale and the fixed effects of the condition and session; individual differences represented by ID as a random effect. The model was specified as:

$$Score[ij] \sim Condition[ij] + Session[ij] + ID[i]$$

The fixed effect of the dog showed a significant intercept estimate of 5.909 ($p < 0.001$). The plant condition had an estimate of −2.588 ($p < 0.001$), and the replica condition an estimate of −2.405 ($p < 0.001$), both compared to the dog condition. The session variable also exhibited a significant effect, with an estimated coefficient of −0.180 ($p = 0.014$). The outcomes of the model are shown in Table 7.

The pairwise comparisons of both dog vs. plant (estimate = 2.588, $p < 0.0001$, Cohen's $d = 2.704$) and dog vs. replica (estimate = 2.405, $p < 0.0001$, Cohen's $d = 2.513$) were significant. The plant condition was not significantly different from the replica for the interest subscale (estimate = −0.182, $p = 0.6296$, Cohen's $d = -0.191$). Detailed pairwise comparisons are shown in S4 Table.

**Value.** An LMM was employed to investigate the relationship between the dependent variable of the IMI score for the value subscale and the fixed effects of the condition and session; individual differences represented by ID were accounted for as a random effect. The model was specified as:

$$Score[ij] \sim Condition[ij] + Session[ij] + ID[i]$$

**Table 6. Descriptive analysis of IMI questionnaire.**

| Condition | Subscale | Mean | Median | SD | Min | Max | Count |
|---|---|---|---|---|---|---|---|
| Dog | choice | 5.599 | 6.000 | 1.233 | 1.625 | 7.000 | 87 |
| | interest | 5.550 | 5.875 | 1.016 | 2.625 | 7.000 | 87 |
| | value | 5.061 | 5.222 | 1.171 | 1.667 | 7.000 | 87 |
| Plant | choice | 4.855 | 5.000 | 1.689 | 1.500 | 7.000 | 87 |
| | interest | 2.963 | 2.875 | 1.019 | 1.125 | 5.625 | 87 |
| | value | 2.963 | 2.875 | 1.019 | 1.125 | 5.625 | 87 |
| Replica | choice | 4.973 | 5.250 | 1.596 | 1.250 | 7.000 | 87 |
| | interest | 3.145 | 3.125 | 1.370 | 1.000 | 6.750 | 87 |
| | value | 2.844 | 2.667 | 1.376 | 1.000 | 6.444 | 87 |

*Note. Mean, median, standard deviation (SD), minimum and maximum of Intrinsic Motivation Inventory (IMI) scores for each condition and each subscale.*

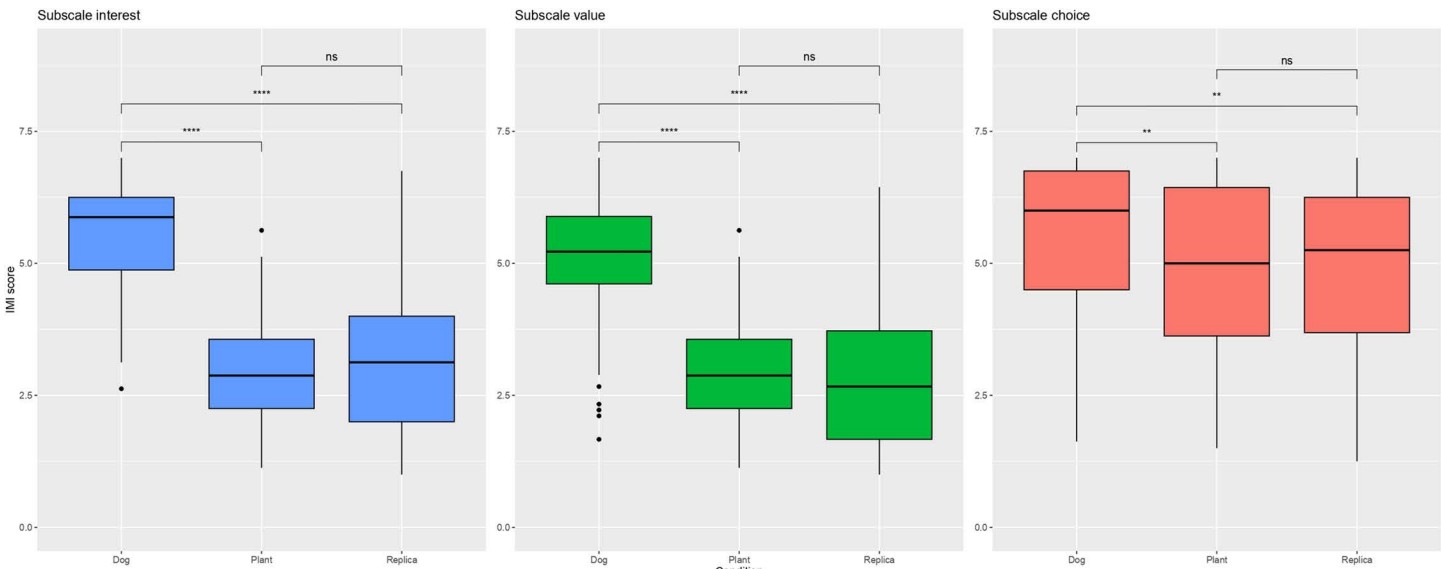

**Fig 4. Boxplot of Intrinsic Motivation Inventory (IMI) scores.** Left panel: subscale interest. Middle panel: subscale value. Right panel: subscale choice. Score across the Dog, Plant, and Replica conditions. Each box shows the interquartile range (IQR) with the median line, and whiskers extend to 1.5×IQR. N=29, repeated measurements over three sessions. Significance level: 0 to 0.0001: ****, >0.0001 to 0.001: ***, >0.001 to 0.01: **, >0.01 to 0.05: *, >0.05: ns (not significant).

For the fixed effect of the dog, the intercept was estimated to be 5.136 ($p<0.001$). The factor condition showed significant coefficients for each level. The plant had an estimated coefficient of −2.099 ($p<0.001$), which was significantly different from the dog condition. The replica condition had an estimated coefficient of −2.217 ($p<0.001$), also significantly different from the dog condition. The variable of the session did not exhibit significance. The outcomes of the model are shown in Table 8.

Both pairwise comparisons, dog vs. plant (estimate=2.099, $p<0.0001$, Cohen's $d=2.175$) and dog vs. replica (estimate=2.217, $p<0.0001$, Cohen's $d=2.298$), showed significance. However, there was no significant difference between the plant and replica conditions (estimate=0.118, $p=1.000$, Cohen's $d=0.123$). Detailed pairwise comparisons are shown in S5 Table.

**Choice.** An LMM was used to examine the association between the dependent variable of the IMI score for the choice subscale and the fixed effects of the condition and session; individual differences represented by ID were accounted for as random effects. The model was specified as:

$$Score[ij] \sim Condition[ij] + Session[ij] + ID[i]$$

**Table 7. Results for condition effects for IMI subscale interest.**

| Predictors | Estimate | SE | CI_Lower | CI_Upper | p_value |
|---|---|---|---|---|---|
| Intercept | 5.909 | 0.212 | 5.495 | 6.324 | <0.001 |
| Condition Plant | −2.588 | 0.145 | −2.871 | −2.304 | <0.001 |
| Condition Replica | −2.405 | 0.145 | −2.689 | −2.121 | <0.001 |
| Session | −0.180 | 0.073 | −0.321 | −0.038 | 0.014 |

*Note. Estimates, standard errors (SE), confidence intervals (CI), and p-values are presented for each predictor. Confidence intervals represent the 95% confidence level.*

**Table 8. Results for condition effects for IMI subscale value.**

| Predictors | Estimate | SE | CI_Lower | CI_Upper | p_value |
|---|---|---|---|---|---|
| Intercept | 5.136 | 0.224 | 4.699 | 5.572 | <0.001 |
| Condition Plant | −2.099 | 0.146 | −2.385 | −1.813 | <0.001 |
| Condition Replica | −2.217 | 0.146 | −2.503 | −1.931 | <0.001 |
| Session | −0.037 | 0.073 | −0.180 | 0.106 | 0.612 |

*Note. Estimates, standard errors (SE), confidence intervals (CI), and p-values are presented for each predictor. Confidence intervals represent the 95% confidence level.*

In terms of fixed effects, the intercept of the dog condition was estimated at 5.856 ($p$<0.001). The plant and replica-dog conditions had respective estimated coefficients of −0.744 ($p$<0.001) and −0.626 ($p$<0.001). Additionally, the variable of the session exhibited significance, with an estimated coefficient of −0.129 ($p$=0.035). The outcomes of the model are shown in Table 9.

Pairwise comparisons revealed significant differences between the dog and the plant (estimate=0.744, $p$<0.0001, Cohen's $d$=0.928) and between the dog and the replica (estimate=0.626, $p$<0.0001, Cohen's $d$=0.781). However, no significant difference was observed between the plant and the replica conditions (estimate=−0.118, $p$=1.000, Cohen's $d$=−0.147). Detailed pairwise comparisons are shown in S6 Table.

**MDWB.** Descriptive MDWB. A descriptive analysis of the MDWB questionnaire can be found in Table 10 and a boxplot of the distribution of MDWB scores for each subscale in Fig 5.

**GB.** An LMM was applied to explore the association between the dependent variable of the GB score and the fixed effects of the condition and session; individual differences represented by ID were accounted for as random effects. The final model was specified as:

$$Score[ij] \sim Condition[ij] + Session[ij] + ID[i]$$

The dog condition had an estimated coefficient of 0.655 ($p$<0.001) showing significant effect on the score. The plant condition as well as the replica condition did not have a significant impact on the score. The variable of the session had an estimated coefficient of −0.185 ($p$=0.014). The outcomes of the model are shown in Table 11.

The pairwise comparisons between the conditions showed significant differences between the baseline and the dog condition (estimate=−0.655, $p$=0.0012, Cohen's $d$=−0.572), the dog and plant conditions (estimate=0.862, $p$<0.001, Cohen's $d$=0.752), and the dog and replica conditions (estimate=0.759, $p$<0.001, Cohen's $d$=0.662). No significant differences were measured for the other pairwise comparisons. Detailed pairwise comparisons are shown in S7 Table.

**Table 9. Results for condition effects for IMI subscale choice.**

| Predictors | Estimate | SE | CI_Lower | CI_Upper | p_value |
|---|---|---|---|---|---|
| Intercept | 5.856 | 0.284 | 5.297 | 6.416 | <0.001 |
| Condition Plant | −0.744 | 0.122 | −0.982 | −0.506 | <0.001 |
| Condition Replica | −0.626 | 0.122 | −0.864 | −0.389 | <0.001 |
| Session | −0.129 | 0.061 | −0.247 | −0.010 | 0.036 |

*Note. Estimates, standard errors (SE), confidence intervals (CI), and p-values are presented for each predictor. Confidence intervals represent the 95% confidence level.*

**Table 10. Descriptive analysis of the MDWB questionnaire.**

| Condition | Subscale | Mean | Median | SD | Min | Max | Count |
|---|---|---|---|---|---|---|---|
| Baseline | GB | 17.724 | 18 | 1.703 | 13 | 20 | 87 |
| | CR | 16.943 | 17 | 2.340 | 11 | 20 | 87 |
| | WT | 14.207 | 15 | 2.922 | 7 | 20 | 87 |
| Dog | GB | 18.379 | 18 | 1.457 | 13 | 20 | 87 |
| | CR | 17.977 | 18 | 1.823 | 13 | 20 | 87 |
| | WT | 15.161 | 15 | 3.000 | 7 | 20 | 87 |
| Plant | GB | 17.517 | 18 | 1.711 | 13 | 20 | 87 |
| | CR | 17.517 | 18 | 2.161 | 12 | 20 | 87 |
| | WT | 12.701 | 13 | 3.069 | 5 | 20 | 87 |
| Replica | GB | 17.621 | 18 | 1.793 | 12 | 20 | 87 |
| | CR | 17.253 | 18 | 2.174 | 11 | 20 | 87 |
| | WT | 13.816 | 14 | 2.683 | 8 | 20 | 87 |

*Note. Mean, median, standard deviation (SD), Minimum and Maximum of multidimension well-being (MDWB) scores for each condition and each sub-scale. GB = good–bad mood; WT = wakefulness–tiredness; CR = calmness–restlessness*

**WT.** An LMM was applied to explore the association between the dependent variable of the WT score and the fixed effects of the condition and session; individual differences represented by ID were accounted for as random effects. The final model was specified as:

$$Score[ij] \sim Condition[ij] + Session[ij] + ID[i]$$

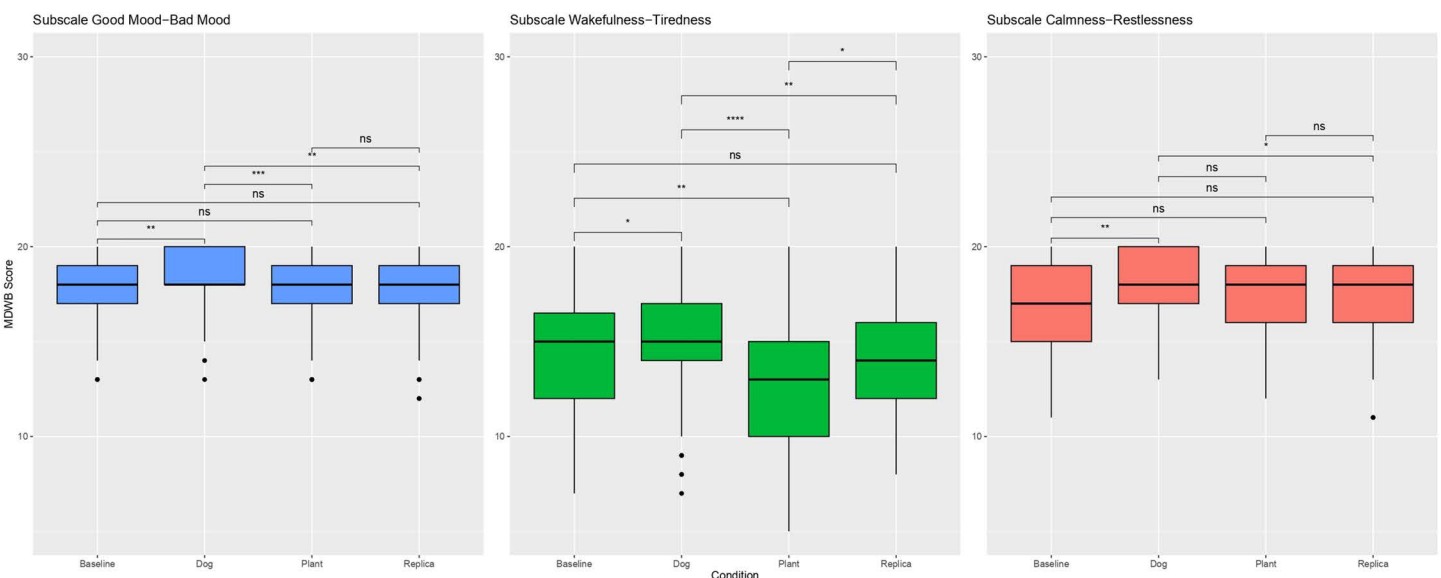

**Fig 5. Boxplot of Multidimensional Wellbeing Questionnaire (MDWB) scores.** Left panel: subscale good mood–bad mood. Middle panel: subscale wakefulness–tiredness. Right panel: Subscale calmness–restlessness. Score across the Dog, Plant, Replica and Baseline conditions. Each box shows the interquartile range (IQR) with the median line, and whiskers extend to 1.5 × IQR. N = 29, repeated measurements over three sessions. Significance level: 0 to 0.0001: ****, > 0.0001 to 0.001: ***, > 0.001 to 0.01: **, > 0.01 to 0.05: *, > 0.05: ns (not significant).

**Table 11. Results for condition effects for MDWB subscale GB.**

| Predictors | Estimate | SE | CI_Lower | CI_Upper | p_value |
|---|---|---|---|---|---|
| Intercept | 18.095 | 0.299 | 17.509 | 18.680 | <0.001 |
| Condition Dog | 0.655 | 0.174 | 0.316 | 0.995 | <0.001 |
| Condition Plant | −0.207 | 0.174 | −0.546 | 0.133 | 0.235 |
| Condition Replica | −0.103 | 0.174 | −0.443 | 0.236 | 0.552 |
| Session | −0.185 | 0.075 | −0.332 | −0.038 | 0.014 |

*Note. Estimates, standard errors (SE), confidence intervals (CI), and p-values are presented for each predictor. Confidence intervals represent the 95% confidence level.*

The effects of the individual conditions show that the dog condition had a significant effect with an estimate of 0.954 ($p = 0.004$). For the plant condition a significant effect was also observed with an estimated coefficient of −1.506 ($p < 0.001$) The replica as well as the session variable are not significant. The outcomes of the model are shown in Table 12.

The significant differences after the pairwise calculations were baseline vs. dog (estimate = −0.954, $p = 0.0246$, Cohen's $d = −0.439$), baseline vs. plant (estimate = 1.506, $p < 0.0001$, Cohen's $d = 0.694$), dog vs. plant (estimate = 2.460, $p < 0.0001$, Cohen's $d = 1.133$), dog vs. replica (estimate = 1.345, $p = 0.0003$, Cohen's $d = 0.620$), and plant vs. replica (estimate = −1.115, $p = 0.0049$, Cohen's $d = −0.514$). The other comparisons did not show significant differences. Detailed pairwise comparisons are shown in S8 Table.

**CR.** An LMM analysis was conducted to explore the association between the dependent variable of the CR score and the fixed effects of the condition and session; individual differences represented by ID were accounted for as random effects. The final model was specified as:

$$Score[ij] \sim Condition[ij] + Session[ij] + + ID[i]$$

For the effects of the conditions are significant for the dog with an estimated coefficient of 1.034 ($p < 0.001$) and the plant with an estimated coefficient of 0.575 ($p = 0.011$). The replica variable had no significant effect. The session seems to influence the model with an estimate of −0.203 ($p = 0.039$). The outcomes of the model are shown in Table 13.

The pairwise comparisons revealed significant differences for two comparisons, which were baseline vs. dog (estimate = −1.034, $p < 0.001$, Cohen's $d = −0.696$) and dog vs. replica (estimate = 0.724, $p = 0.0087$, Cohen's $d = 0.487$). Detailed pairwise comparisons are shown in S9 Table.

**Table 12. Results for condition effects for MDWB subscale WT.**

| Predictors | Estimate | SE | CI_Lower | CI_Upper | p_value |
|---|---|---|---|---|---|
| Intercept | 13.750 | 0.519 | 12.734 | 14.766 | <0.001 |
| Condition Dog | 0.954 | 0.329 | 0.311 | 1.600 | 0.004 |
| Condition Plant | −1.506 | 0.329 | −2.149 | −0.863 | <0.001 |
| Condition Replica | −0.391 | 0.329 | −1.034 | 0.252 | 0.236 |
| Session | 0.2284 | 0.1425 | −0.050 | 0.507 | 0.110 |

*Note. Estimates, standard errors (SE), confidence intervals (CI), and p-values are presented for each predictor. Confidence intervals represent the 95% confidence level.*

**Table 13. Results for Condition Effects for MDWB Subscale CR.**

| Predictors | Estimate | SE | CI_Lower | CI_Upper | p_value |
|---|---|---|---|---|---|
| Intercept | 17.348 | 0.381 | 16.601 | 18.095 | <0.001 |
| Condition Dog | 1.034 | 0.225 | 0.594 | 1.475 | <0.001 |
| Condition Plant | 0.575 | 0.225 | 0.134 | 1.015 | 0.011 |
| Condition Replica | 0.310 | 0.225 | −0.130 | 0.751 | 0.169 |
| Session | −0.203 | 0.098 | −0.393 | −0.012 | 0.039 |

*Note. Estimates, standard errors (SE), confidence intervals (CI), and p-values are presented for each predictor. Confidence intervals represent the 95% confidence level.*

## Discussion

This study aimed to assess whether interacting with a real dog, compared to a replica or a plant, would increase approach motivation and positive affect in healthy adults, using both physiological (FAA) and subjective (IMI, MDWB) measures. While FAA did not differ across conditions, self-reported measures revealed clear benefits of interacting with a real dog.

Participants consistently reported higher interest, value and perceived choice after interacting with the dog than with the control stimuli as shown by the IMI. Similarly, MDWB scores showed that participants had better mood, higher alertness, and greater calmness after interacting with a real dog compared to the controls. These findings align with previous research [9,11] and highlight the subjective value of real HAIs. The real dog also significantly enhanced mood, alertness, and calmness (MDWB) compared to the control stimuli, supporting earlier findings from Yoo et al. [8] and reinforcing the notion that brief interactions with dogs can enhance emotional well-being.

However, contrary to our hypothesis, FAA did not show condition-related differences. This discrepancy may be due to our brief intervention design. Prior studies [31,47] reported FAA changes after extended interventions (8–10 weeks), suggesting that brief or one-time interactions may be insufficient to elicit measurable neural changes due to their lack of longitudinal exposure. This interpretation is supported by Barcelona et al. [30], who also failed to find FAA differences across brief interventions. They also noted that FAA increases with age and potentially with repeated exposure. Given that our participants were relatively young (mean age: 28, median age: 25), age-related FAA effects were likely minimal because asymmetry increases with age as measured by Barros et al. [48] where young adults (18–34 y.o.) showed smaller FAA as older adults. Importantly, while FAA is a popular marker of approach motivation, it often fails to correlate with subjective reports [49,50]. Such dissociation might be due to temporal mismatches between neurophysiological and behavioral/emotional changes [51].

One theoretical framework that may explain the stronger subjective responses to the real dog is the concept of *biophilia* that hypothesizes an innate human tendency to connect with living beings [52]. While both plants and animals can support emotional regulation [53,54], dogs may mainly trigger social-emotional responses such as joy and attachment [55]. These mechanisms likely underlie the increased mood and motivation scores in our dog condition. Supporting this, Stevens et al. [56] found that dogs can enhance affective states similarly to nature exposure but not cognitive performances, indicating that the effect of interacting with a dog may rely on different cognitive pathways as the effect of interacting with a plant.

Interestingly, despite the zoomorphic appearance of the replica dog, participants did not exhibit the same motivation or emotional responses, suggesting that perceived "aliveness" plays a key role in HAI effects [55]. This distinction may become more pronounced over repeated interactions, as emotional bonding is more likely to develop with real animals over time.

Overall, our results support the idea that repeated or prolonged exposure to real animals may be necessary to observe neural changes, even when subjective benefits are immediately felt. Studies employing a more longitudinal design with prolonged interventions are therefore needed to track potential cumulative effects on both emotional well-being and brain activity.

**Strengths and limitations**

A key strength of this study was its within-subject design, which improved the sensitivity of both behavioral and EEG measures. The sample size, with a total of 428 recordings, allowed for a substantial number of EEG trials across conditions, and the randomized session order helped control for order and habituation effects. The use of both subjective and physiological outcomes provided a multidimensional perspective on participants' responses. Additionally, the subjective measures of motivation were performed based on a congruent action and not based on an image or scenario evaluation. Indeed, Uusberg et al. [57] argued that directly experiencing action-related motivation could have a different impact on frontal asymmetry than a subjective assessment of an image or scenario-related motivation.

However, relying solely on FAA as an objective marker is a limitation. FAA captures only one aspect of motivational neurophysiology and may lack sensitivity in short interventions. Additionally, participants' awareness of the conditions could not be blinded due to the nature of the stimuli. Finally, some questionnaire constructs did not fully match the theoretical constructs associated with FAA, potentially complicating comparisons. Indeed, the IMI is a measure of intrinsic motivation, while FAA is an indicator of approach motivation. Both are closely related concepts, but they are not exactly the same. Cury et al. [58] showed that intrinsic motivation was more important in an approach paradigm than in an avoidance paradigm, demonstrating the close link between the concepts of approach and intrinsic motivation. Similarly, the MDWB questionnaire is not designed to assess positive affect specifically. Components of positive affect, such as positive mood, joy, or alertness [57], are, however, evaluated by the MDWB questionnaire.

**Future research and clinical applications**

Future research should explore how repeated HAIs affect neural markers of motivation and emotion. Incorporating longitudinal designs could determine whether extended exposure to dogs elicits sustained FAA or other physiological changes. Broader neural markers—such as beta activity or regions like the anterior cingulate cortex—should also be investigated using multimodal imaging (e.g., fMRI, EEG).

From a clinical perspective, understanding the neurobiological mechanisms underlying HAI may inform therapy design. As motivation and positive affect are critical in rehabilitation outcomes [59,60], integrating real animals into treatment settings may enhance engagement and psychological benefits. Neurophysiological assessments could help tailor interventions and monitor their effectiveness over time.

**Conclusion**

The present study found that healthy participants did not exhibit different frontal brain activity, measured as FAA, during the presence of a real dog compared to a replica dog or a plant. This electrophysiological measurement suggests that the presence of the real dog had no effect on motivation. However, we did observe a higher self-reported intrinsic motivation and a more positive mental state after the interaction with the real dog compared to the conditions with the replica dog or the plant. These results suggest that processes like motivation and positive affect are underlying mechanisms of AAIs. Moreover, the dissociation between the biological and the psychological outcomes indicates that further research is needed to better understand how HAIs affect brain activity.

**Supporting information**

**S1 Table. Pairwise comparisons for electrode pair Fp1–Fp2.**
(ZIP)

**S2 Table: Pairwise comparisons for electrode pair F3–F4.**
(ZIP)

**S3 Table: Pairwise comparisons for electrode pair F7–F8.**
(ZIP)

**S4 Table. Pairwise comparisons and effect sizes for interest subscale.**
(ZIP)

**S5 Table. Pairwise comparisons and effect sizes for value subscale.**
(ZIP)

**S6 Table. Pairwise comparisons and effect sizes for choice subscale.**
(ZIP)

**S7 Table. Pairwise comparisons and effect sizes for GB subscale.**
(ZIP)

**S8 Table. Pairwise comparisons and effect sizes for WT subscale.**
(ZIP)

**S9 Table. Pairwise comparisons and effect sizes for CR subscale.**
(ZIP)

**S1 Fig. Distribution of frontal alpha asymmetry (FAA) scores for electrode pair Fp1-Fp2.** For each of the four experimental conditions: Baseline 1, Dog, Replica, and Plant. Each point represents an individual participant's FAA score at a given session. The plot illustrates intra-individual variability and potential condition-related trends across sessions. Participant 11 was excluded and is therefore not illustrated in the plot.
(ZIP)

**S2 Fig. Distribution of frontal alpha asymmetry (FAA) scores for electrode pair F3-F4.** For each of the four experimental conditions: Baseline 1, Dog, Replica, and Plant. Each point represents an individual participant's FAA score at a given session. The plot illustrates intra-individual variability and potential condition-related trends across sessions. Participant 11 was excluded and is therefore not illustrated in the plot.
(ZIP)

**S3 Fig. Distribution of frontal alpha asymmetry (FAA) scores for electrode pair F7-F8.** For each of the four experimental conditions: Baseline 1, Dog, Replica, and Plant. Each point represents an individual participant's FAA score at a given session. The plot illustrates intra-individual variability and potential condition-related trends across sessions. Participant 11 was excluded and is therefore not illustrated in the plot.
(ZIP)

**S4 Fig. Correlation of FAA scores for electrode pair Fp1–Fp2 and IMI scores for each condition and subscale.**
FAA = frontal alpha asymmetry IMI = intrinsic motivation inventory; r = Pearson r.
(ZIP)

**S5 Fig. Correlation of FAA scores for electrode pair F3–F4 and IMI scores for each condition and subscale.**
FAA = frontal alpha asymmetry; IMI = intrinsic motivation inventory; r = Pearson r.
(ZIP)

**S6 Fig. Correlation of FAA scores for electrode pair F7–F8 and IMI scores for each condition and subscale.**
FAA = frontal alpha asymmetry; IMI = intrinsic motivation inventory; r = Pearson r.
(ZIP)

**S10 Table. Correlations between interaction time of participants and dog in each session with FAA score for each electrode pair.**
(ZIP)

## Acknowledgments

We would like to thank the Faculty of Psychology at the University of Basel for providing us with a laboratory and EEG equipment. We would also like to thank the dogs involved in this study—Runa, Merlin, and Clay—and their respective handlers: Wanda Arnskötter, Jay Mazumdar, and Eve Gerber.

## Author contributions

**Conceptualization:** Fabio Carbone, Karin Hediger.

**Data curation:** Fabio Carbone, Eve-Yaël Gerber, Camille Rérat.

**Formal analysis:** Fabio Carbone, Eve-Yaël Gerber, Camille Rérat, Jan Hattendorf.

**Funding acquisition:** Karin Hediger.

**Investigation:** Fabio Carbone, Eve-Yaël Gerber, Camille Rérat.

**Methodology:** Fabio Carbone, Jan Hattendorf, Karin Hediger.

**Project administration:** Fabio Carbone, Eve-Yaël Gerber, Camille Rérat, Karin Hediger.

**Resources:** Fabio Carbone, Jan Hattendorf, Karin Hediger.

**Software:** Fabio Carbone, Jan Hattendorf.

**Supervision:** Fabio Carbone, Karin Hediger.

**Visualization:** Fabio Carbone.

**Writing – original draft:** Fabio Carbone.

**Writing – review & editing:** Fabio Carbone, Eve-Yaël Gerber, Camille Rérat, Jan Hattendorf, Karin Hediger.

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
