## [Decision Letter · Decision Letter 0]

7 Mar 2025

Dear Dr. Fabio Carbone,

Thank you for submitting your manuscript to PLOS ONE. After careful consideration, we feel that it has merit but does not fully meet PLOS ONE’s publication criteria as it currently stands. Therefore, we invite you to submit a revised version of the manuscript that addresses the points raised during the review process.

We look forward to receiving your revised manuscript.

Kind regards,

Imtiaz Hussain, Ph.D

Academic Editor

PLOS ONE

 [Funding for this project was provided by the Swiss National Science Foundation (SNSF) through the Eccellenza grant PCEFP1_194591 / 1.]. 

3. In the online submission form, you indicated that [All data and codes are available here: https://osf.io/tz26c/?view_only=d673459b9ad64b6986173aa97d129b21 or upon reasonable requests.].

Reviewers' comments:

Reviewer's Responses to Questions

**Comments to the Author**

1. Is the manuscript technically sound, and do the data support the conclusions?

Reviewer #1: Partly

Reviewer #2: Partly

2. Has the statistical analysis been performed appropriately and rigorously?

Reviewer #1: Yes

Reviewer #2: No

3. Have the authors made all data underlying the findings in their manuscript fully available?

Reviewer #1: Yes

Reviewer #2: Yes

4. Is the manuscript presented in an intelligible fashion and written in standard English?

Reviewer #1: Yes

Reviewer #2: Yes

Reviewer #1: Carbone et al. investigated the neurological aspects of human-animal interactions (HAIs) by exploring frontal alpha asymmetry (FAA) to understand how humans respond to animal interactions. The authors experimented with 29 adults who interacted with a real dog, a replica dog, and a plant. The authors measured FAA as an objective marker of neural brain activity, and they also assessed subjective experiences of motivation and positive affect via established questionnaires. It was noted that while objective measurements (FAA) did not show significant differences, the subjective assessments revealed higher motivation and a more positive state of mind after interacting with a real dog. Overall, this study is interesting. The manuscript is well-written and sufficiently rigorous in most places, but several points need to be clarified throughout the manuscript.

Please consider the following suggestions for improving the manuscript:

Comment #1: Abstract: My suggestion for improvement might be to add the potential practical applications of their findings in real-world contexts. Specifically, this study showed that there is the possibility that human-animal interactions may not engage frontal brain regions in the way expected.

Comment #2: The introduction is long. Please shorten and organize it. It could benefit from a more structured flow. For example, It could start with a general context about human-animal interactions that can improve mood and motivation, followed by specific gaps in the literature, and next, the rationale behind choosing FAA as a well-established neural correlate of motivation and positive affect. Results from previous studies need to be summarized as the foundation for the current research (e.g., lines 95-129) and whether these studies are part of animal therapy with post-sickness or with healthy participants. Finally, instead of introducing hypotheses in different places (lines 133, 147, 149), the authors should put forward clear objectives and hypotheses at the end of the introduction and how they carried out experiments to verify these hypotheses and highlight the key findings of this study.

Comment #3: Data analysis related (lines 345-379): Since you used lme4 packages for linear mixed-effect model, which is a valid choice and necessary. However, Did you consider initially using repeated-measures ANOVA with an appropriate posthoc test (as an extra exploration, not for visualization), especially since each participant experienced three sessions with all conditions to link/pair the results from each treatment/session to the same participant for the pairwise comparisons as shown in your supplementary tables? Moreover, I missed information about how the repeated measures of the 90 sessions (3 replicates for each participant) were averaged and varied in three trials for each participant within three consecutive weeks.

Comment #4: While you claimed that one female participant was excluded from the study (line 408), I see the characteristics for thirty participants in table 1 (lines 412-413). Which one was excluded? It is probably better to delete this row if it does not fit your criteria, as stated in the method section.

Comment #5: Age-related (line 407-414): min = 19 years and max = 62 years; I have a concern about the young adult stated in line 407. Ages like 45, 52, and 63 are not typically young unless your study defines young adults differently. Part of your discussion (lines 696-699) claims that the changes were not measurable between participants due to their young age (28.07 years). Please clarify!

Comment #6: Figures 3-5-related. Since the recordings and measurements could vary among individuals, it would be nice to show all the data points of individual values behind means (n=29 X 3 replicates sessions or average) as scatter plots instead of boxplots to actually show how the measurements are distributed among participants and sessions, especially with many outliers, considering you have only 29 participants. Additionally, the figure legends contain very limited information. I would rather expand some information about the abbreviations, statistical tests used, sample size, plot type, mean or median data, upper and lower whiskers representation,..etc.

Comment #8: Given that the subjective measures indicate significant differences between interactions with real dogs and control conditions, even if these are not reflected in the frontal EEG alpha asymmetry data. I am missing explanations in the discussion on how repeated interactions with animals could affect motivation and emotional well-being. The discussion part here is very long and is not a real discussion; it should be reorganized thoroughly. The first paragraph should address your main results (lines 672-706). Merge these two paragraphs: (lines 677-679) and (lines 764-767). Don't repeat the results and statistics you havep already shown before (e.g., lines 707- 741). Shorten the paragraphs at lines 742-763. Please shorten these subsections: strengths and limitations (lines 794-849) & future research and clinical applications (851-879). For discussion, you should present your main results and compare them with other research, and then discuss the reason why different research shows different results.

Reviewer #2: I have thoroughly seen this article. The idea is unique, but there are some comments which should be considered for screening the article in a good way.

Comments-1 The author must have to provide proper updated reference in many statement the consistent and contrast references are missing which making the study weaker.

Comments-2 Proper procedure of randomization is not followed in the trial.

Comments-3 For trial based study it is mandatory to specify the inclusion and exclusion criteria for the subjects under study in each groups.

Comments-4 Although qualitative and quantitative touch is given in the study but it is not statistically significantly discussed in the abstract of the manuscript.

Comments-5 If the author discussed about intervention in various groups but it not properly discussed in the manuscript and for intervention proper role of randomization is mandatory to avoid biasness. So the author has to scientifically clarify the matter. Thanks

Comments-6 In line number 161 its coated that “Recruitment was conducted via online advertisements” it’s not clear that which online software and AI tool was used for the said procedure.

Comments-7 Line number 233 -236 statement need proper consistent reference to support the current study please.

Comments-8 The author must have to properly explain the type and structure of questionnaires which is implicated in the study.

Comments-9 In line number 331 the author used IMI questionnaire please clear and explain what type and how many variety of questionnaires were used in the trial.

Comments-10 Which specie of the plant you have used in this trial?

Commetns-11 is there any relation of Dogs breeds in this experiment? In my point of view every breed have different characteristics

Comments-12 Which type of stress you have focused in this trail?

**Do you want your identity to be public for this peer review?** For information about this choice, including consent withdrawal, please see our Privacy Policy

Reviewer #1: No

Reviewer #2: No

---

## [Author Response · Author response to Decision Letter 0]

24 Apr 2025

Response letter to reviewers

PONE-D-24-51588

Neuromechanisms and subjective experiences during human-dog interactions: assessing motivation and mental state in a randomized, controlled trial

PLOS ONE

Reviewer 1

Comment #1: Abstract: My suggestion for improvement might be to add the potential practical applications of their findings in real-world contexts. Specifically, this study showed that there is the possibility that human-animal interactions may not engage frontal brain regions in the way expected.

Thank you for this suggestion of improvement. A sentence was added to the abstract to define a potential real-world application. We do not wish to mention that frontal region may not be engaged in HAI as previous studies indicate the contrary and that other findings from our study (soon to be published) show evidence of frontal region implication during HAI.

Comment #2: The introduction is long. Please shorten and organize it. It could benefit from a more structured flow. For example, It could start with a general context about human-animal interactions that can improve mood and motivation, followed by specific gaps in the literature, and next, the rationale behind choosing FAA as a well-established neural correlate of motivation and positive affect. Results from previous studies need to be summarized as the foundation for the current research (e.g., lines 95-129) and whether these studies are part of animal therapy with post-sickness or with healthy participants. Finally, instead of introducing hypotheses in different places (lines 133, 147, 149), the authors should put forward clear objectives and hypotheses at the end of the introduction and how they carried out experiments to verify these hypotheses and highlight the key findings of this study.

Thank you for your constructive remark and helpful suggestions. We acknowledge that the introduction was poorly structured, too long and containing unnecessary information. We rewrote the introduction following your suggestions and made it more concise.

Comment #3: Data analysis related (lines 345-379): Since you used lme4 packages for linear mixed-effect model, which is a valid choice and necessary. However, Did you consider initially using repeated-measures ANOVA with an appropriate posthoc test (as an extra exploration, not for visualization), especially since each participant experienced three sessions with all conditions to link/pair the results from each treatment/session to the same participant for the pairwise comparisons as shown in your supplementary tables? Moreover, I missed information about how the repeated measures of the 90 sessions (3 replicates for each participant) were averaged and varied in three trials for each participant within three consecutive weeks.

Thank you for your comment on the statistical analysis and your suggestion for alternative tests. The statistical plan was discussed and approved together with a statistician. We used a linear mixed-effect model as it works well for within-subject design with repeated measurements and allows some flexibility in the assumptions of data distribution which is less true with a repeated-measures ANOVA. The repetitions of measurements were taken account in the model by having the “sessions” as fixed effect but the score of each session was considered individually and not averaged.

Comment #4: While you claimed that one female participant was excluded from the study (line 408), I see the characteristics for thirty participants in table 1 (lines 412-413). Which one was excluded? It is probably better to delete this row if it does not fit your criteria, as stated in the method section.

Thank you for noticing this discrepancy, indeed this was a mistake to have all 30 screened participants in the table. We marked the participant that was excluded in order to clearly indicate that this person was not included in the analysis.

Comment #5: Age-related (line 407-414): min = 19 years and max = 62 years; I have a concern about the young adult stated in line 407. Ages like 45, 52, and 63 are not typically young unless your study defines young adults differently. Part of your discussion (lines 696-699) claims that the changes were not measurable between participants due to their young age (28.07 years). Please clarify!

Thanks for pointing out this concern. We considered that the averaged aged of all participants could be considered as young even if some participants are older. Indeed, the average age is 28 years and the median is 25 which is within or close to the general definition of young adulthood that is between 18 and 26 up to 30 years old depending on the source. Moreover only 17% of the participants are above this limit of 30 years, we then decided to consider our sample as young. Finally, the reference we used to state that younger adults show smaller FAA than older adults worked with young adults ranging from 18 to 34 years.

Comment #6: Figures 3-5-related. Since the recordings and measurements could vary among individuals, it would be nice to show all the data points of individual values behind means (n=29 X 3 replicates sessions or average) as scatter plots instead of boxplots to actually show how the measurements are distributed among participants and sessions, especially with many outliers, considering you have only 29 participants. Additionally, the figure legends contain very limited information. I would rather expand some information about the abbreviations, statistical tests used, sample size, plot type, mean or median data, upper and lower whiskers representation,..etc.

Thank you for suggesting additional plots. We have created scatter plots showing individual distribution of FAA score among participants in all conditions and sessions for each electrode pair. They are visible in the supplementary material. In addition, captions of figures have been extended and more information has been added.

Comment #8 (there was no comment #7): Given that the subjective measures indicate significant differences between interactions with real dogs and control conditions, even if these are not reflected in the frontal EEG alpha asymmetry data. I am missing explanations in the discussion on how repeated interactions with animals could affect motivation and emotional well-being. The discussion part here is very long and is not a real discussion; it should be reorganized thoroughly. The first paragraph should address your main results (lines 672-706). Merge these two paragraphs: (lines 677-679) and (lines 764-767). Don't repeat the results and statistics you have already shown before (e.g., lines 707- 741). Shorten the paragraphs at lines 742-763. Please shorten these subsections: strengths and limitations (lines 794-849) & future research and clinical applications (851-879). For discussion, you should present your main results and compare them with other research, and then discuss the reason why different research shows different results.

Again, thank you for this comment and the great suggestions of improvement. We agreed with you on most of the points and have modified the discussion accordingly. The structure has been changed to fit better to an academic discussion and the content has been synthetized and rewritten.

Reviewer 2

Comments-1 The author must have to provide proper updated reference in many statement the consistent and contrast references are missing which making the study weaker.

Thank you for rising this problem. The introduction and discussion have been rewritten and the referencing has been refined. If you are referring to any particular statement of the text, please provide precisions.

Comments-2 Proper procedure of randomization is not followed in the trial.

Thank you for highlighting this point. We describe the procedure of randomization that consisted, in a first step, in randomizing the order of exposure of the three experimental conditions for all session. Then three sessions were randomly assigned to each ID number from 1 to 30. Lastly each recruited participant was assigned to an ID number.

Comments-3 For trial based study it is mandatory to specify the inclusion and exclusion criteria for the subjects under study in each group.

Thanks for your comment. Inclusion and exclusion criteria are described in section “Study Population” in lines 112-125. As the design of the study is a within subjected design, there is only one group where all participants undergo the same conditions therefore inclusion and exclusion criteria are the same for each participant and do not depend on a group attribution.

Comments-4 Although qualitative and quantitative touch is given in the study but it is not statistically significantly discussed in the abstract of the manuscript.

Thanks for your suggestion. As the guidelines of the journal state that the results should be summarized in the abstract and not detailed too extensively, the authors made the choice to not include statistics in the abstract as it would include lots of information and might make it less readable and too long.

Comments-5 If the author discussed about intervention in various groups but it not properly discussed in the manuscript and for intervention proper role of randomization is mandatory to avoid biasness. So the author has to scientifically clarify the matter. Thanks

Thank you for noticing this. As explained in comment 3, the study employs a within-subject design meaning that all participants undergo all interventions. Randomization procedure is explained in comment 2.

Comments-6 In line number 161 its coated that “Recruitment was conducted via online advertisements” it’s not clear that which online software and AI tool was used for the said procedure.

Thank you for asking some clarifications on this matter. We have clarified in the manuscript the type of website that was used to advertise the study and to whom it was accessible.

Comments-7 Line number 233 -236 statement need proper consistent reference to support the current study please.

Thank you for asking. As the statement you mention here describe the methodology and a specific action during the session (i.e. monitoring contact duration) we do not think that this particular statement requires a specific reference.

Comments-8 The author must have to properly explain the type and structure of questionnaires which is implicated in the study.

Thank you for your advice. Each questionnaire has been described in detail (number of items, max and min score and their respective meaning, method to calculate final score). Please provide specifications if you are referring to something more precise.

Comments-9 In line number 331 the author used IMI questionnaire please clear and explain what type and how many variety of questionnaires were used in the trial.

Thanks for asking for more precisions. The IMI questionnaire is described in detail and information are given about the type of IMI that was chosen to specifically assess the motivation after an activity. If you are referring to something in particular, please provide us with some details.

Comments-10 Which specie of the plant you have used in this trial?

Thank you for asking. The specie of the plant is indicated the section Material and Methods->Plant (lines 214-215). It is Epipremnum aureum

Comments-11 is there any relation of Dogs breeds in this experiment? In my point of view every breed have different characteristics

Thank you for this question, that is a valid comment. Different breeds can indeed be perceived as aggressive or non-aggressive depending on physical characteristics (Briones et al., 2022). But the three dog we involved in the project have physical properties associated with non-aggressive breeds (in our case Bernese Mountain dog, Australian Shepard and retriever). Moreover, all three dogs were of equivalent size being medium to large size which should limit different perceptions between of the dogs.

Comments-12 Which type of stress you have focused in this trail?

Thank you for asking. The trial did not investigated effect on stress nor intended to do so. Please provide specifications if you are referring to something in particular.

---

## [Decision Letter · Decision Letter 1]

12 May 2025

Neuromechanisms and subjective experiences during human-dog interactions: assessing motivation and mental state in a randomized, controlled trial

PONE-D-24-51588R1

Dear Dr. Fabio Carbone,

We’re pleased to inform you that your manuscript has been judged scientifically suitable for publication and will be formally accepted for publication once it meets all outstanding technical requirements.

Kind regards,

Imtiaz Hussain, Ph.D

Academic Editor

PLOS ONE

Additional Editor Comments (optional):

Reviewers' comments:

Reviewer's Responses to Questions

**Comments to the Author**

Reviewer #1: All comments have been addressed

Reviewer #2: All comments have been addressed

2. Is the manuscript technically sound, and do the data support the conclusions?

Reviewer #1: Yes

Reviewer #2: (No Response)

3. Has the statistical analysis been performed appropriately and rigorously?

Reviewer #1: Yes

Reviewer #2: (No Response)

4. Have the authors made all data underlying the findings in their manuscript fully available?

Reviewer #1: Yes

Reviewer #2: Yes

5. Is the manuscript presented in an intelligible fashion and written in standard English?

Reviewer #1: Yes

Reviewer #2: Yes

Reviewer #1: The authors have significantly improved the manuscript. It is now well-structured and clear, and all my comments have been addressed.

Reviewer #2: It is well enough that you have justify the queries. The questions which i have raised during review, i am glad to see your justification. However, it is my suggestion such studies should have another more clear view so that these sort of issues may provide a line for other researchers as well.

**Do you want your identity to be public for this peer review?** For information about this choice, including consent withdrawal, please see our Privacy Policy

Reviewer #1: No

Reviewer #2: No

---

## [Editor Report · Acceptance letter]

PONE-D-24-51588R1

PLOS ONE

Dear Dr. Carbone,

I'm pleased to inform you that your manuscript has been deemed suitable for publication in PLOS ONE. Congratulations! Your manuscript is now being handed over to our production team.

Kind regards,

on behalf of

Dr. Imtiaz Hussain

Academic Editor

PLOS ONE